# Evaluation of Three-Dimensional Surface Roughness in Microgroove Based on Bidimensional Empirical Mode Decomposition

**DOI:** 10.3390/mi13112011

**Published:** 2022-11-18

**Authors:** Haiyu Jiang, Wenqin Li, Zhanjiang Yu, Huadong Yu, Jinkai Xu, Lei Feng

**Affiliations:** National and Local Joint Engineering Laboratory for Precision Manufacturing and Detection Technology, Changchun University of Science and Technology, Changchun 130012, China

**Keywords:** non-Gaussian surface, curved surface of the microgroove, three-dimensional surface roughness, bidimensional empirical mode decomposition, extraction of the reference plane

## Abstract

Micromilling is an extremely important advanced manufacturing technology in the micromanufacturing industry. Compared with the traditional milling process, micromilling has stricter requirements on the surface roughness of the workpiece, and the roughness of the microcurved surface is not easy to measure. In order to more accurately characterize the curved surface morphology of the microgrooves obtained by micromilling, this paper proposes a method to extract the reference plane of the curved surface based on the bidimensional empirical mode decomposition algorithm and characterize the three-dimensional surface roughness of the curved surface. First, we synthesize the morphologies of the microgrooves by simulated non-Gaussian rough surfaces and models of textures. Second, the bidimensional empirical mode decomposition algorithm was used to extract the reference planes of the simulated microgrooves. Third, the three-dimensional roughness parameters suitable for the curved surfaces of microgrooves were selected to establish an evaluation system. The results show that the mean squared errors of the reference planes are below 1%, so bidimensional empirical mode decomposition can effectively extract reference planes, and the evaluation system of three-dimensional surface roughness proposed in this paper reflects morphological characteristics of the curved surfaces of microgrooves more thoroughly than that of two-dimensional surface roughness parameters.

## 1. Introduction

Micromilling is widely used in aerospace, modern medicine, architecture, and microelectronics because of its advantages of high processing precision, high processing efficiency, and ability to process three-dimensional precision microparts [1,2,3,4]. Compared with conventional milling, micromilling has more stringent requirements for surface quality. The roughness of the micromilling surface directly affects the wear resistance, the sealing efficiency, and the corrosion resistance of parts, which is one of the essential indicators to measure the quality of micromilling [5,6,7]. The characterization of two-dimensional surface roughness has been widely used in scientific analysis and engineering practice; however, its characterization parameters can no longer meet the needs of technological development for the evaluation of surface topography information. In contrast, as the three-dimensional surface roughness can reflect the surface state of the workpiece more comprehensively, it has become a research hotspot [8,9]. In recent years, researchers have mostly focused on the characterization of the three-dimensional surface roughness of the planar workpiece, ignoring that of the tiny, curved workpiece. In addition, during the actual measurement of the surface topography of tiny parts, the waviness and shape errors of the surface are prone to be calculated as roughness components. This paper, therefore, studied the three-dimensional surface roughness measurement of the curved surface morphologies of microgrooves machined by micromilling.

The core of three-dimensional surface roughness characterization is the determination of the reference plane. To achieve this, the main methods include a polynomial fitting method, filtering method, Motif method, and fractal method [10,11,12]. Reference [13] presented a new three-dimensional Motif method for multiscale analysis of surface features and proposed a new three-dimensional model to calculate the statistical parameters for surface features. In ref. [14], the fractal dimension was utilized to quantify the feature separation index in the wavelet transform to obtain the composite features of the engineering surface and verify the effectiveness of the method. In ref. [15], wavelet packet transform was used to separate the surface textures, and the reconstructed roughness and waviness coincided well with the original ones, proving that the extracted textures could better reflect the surface structure. However, the polynomial fitting method obtains the polynomial function according to the principle of least squares as the reference of surface roughness evaluation, the reference plane cannot reflect the existence of the secondary structure, resulting in distortion of the fitting result, so it is not suitable for the evaluation of surface roughness in micromilling. The Motif method uses different thresholds to separate the information about the roughness of surface topography, which lacks technical support and reasonable standards. The fractal method can theoretically use fractal dimensions and length of features to reflect the essential features of surface topography, but the finer the surface structure, the more fractal dimensions are required, and the microgroove surface is small and fine enough, so it is not simple enough. The filtering method separates the signal to extract a filtering reference, mainly including Gaussian filtering and wavelet filtering. Gaussian filtering, however, is only suitable for surface topography obeying normal distribution; the actual engineering surface is complicated, which narrows its scope of application. The wavelet basis restricts the effectiveness of wavelet filtering, and the number of decompositions needs to be determined according to the specific situation, resulting in adverse effects on its development.

Nunes J C et al. [16] extended the empirical mode decomposition (EMD) proposed by N.E Hung et al. [17] to two-dimensional signals and proposed bidimensional empirical mode decomposition (BEMD). As an adaptive multiscale multiresolution analysis method of the image, BEMD can decompose nonlinear and nonstationary image signals into a series of bidimensional intrinsic mode functions (BIMF) from high frequency to low frequency. Reference [18] introduced a similarity measure; the noise-compressed image was decomposed into multiple BIMFs by the BEMD algorithm. The soft-interval threshold technique was used to remove the noise of noise-dominated BIMFs and reconstruct the denoised image, providing better performance. Reference [19] constructed a cascade feedback compression framework based on the spatial similarity property of the BIMFs decomposed by BEMD at different resolutions. In ref. [20], BEMD was used to decompose the reference image and the distorted image, the reference image was modeled to measure the damage of the distorted image, and the image quality was evaluated. The roughness information of the microgroove surface is a nonlinear nonstationary signal. A series of surfaces with different but similar scales of surface structure is obtained by BEMD. The adaptability of the scale division avoids the problem of numerical unity of the surface. At the same time, BEMD can eliminate the malformed wave and make each BIMF show a certain periodicity, which is consistent with the characteristics of the actual surface profile. Therefore, the BEMD algorithm is superior to other algorithms in the analysis of surface roughness.

The remainder of this article is organized as follows. In this paper, a rough surface of micromilling was established using the height distribution function and autocorrelation function in Section 2. At the same time, the digital models of different microgroove textures were constructed according to different micromilling parameters, and then the rough surface and texture models were synthesized to acquire different simulated microgroove surfaces. In Section 3, we arranged the height information of the micromilling surface in a matrix and converted it into a two-dimensional image signal, using the BEMD algorithm to extract the reference planes of the microgroove surfaces under different processing parameters, and the error analyses were carried out. In Section 4, we carried out experiments to analyze the surface morphologies of microgrooves and selected the appropriate three-dimensional surface roughness parameters for evaluation. The analysis of the experimental results shows that the reference planes extracted by the BEMD algorithm are suitable for the calculation of the three-dimensional surface roughness of the microgrooves, and the evaluation system of three-dimensional surface roughness can reflect the surface characteristics of the microgrooves more comprehensively.

## 2. The Profile Curves of the Polishing Tool

The topography of the machined surface is an important measure to evaluate the machined surface quality. There are a variety of uncertainties in the actual machining process, so the micromilling surface is simulated with a non-Gaussian rough surface.

### 2.1. The Profile Curves of the Polishing Tool

In this paper, the Johnson conversion system was used to convert Gaussian random sequences into non-Gaussian random sequences with positive skewness and kurtosis, which can generate the required height distribution points well within the height simulation range of rough surfaces. The system frequency curve described by the conversion system is transformed for different situations, including the following three forms [21]:

(1) Unbounded System *S_U_*:(1)$$\eta =\gamma +\delta \;\sin^{-1}\left(\frac{\eta {}^{\prime }-\xi }{\lambda }\right)$$

(2) Lognormal System *S_L_*:(2)$$\eta =\gamma +\delta \;\log\left(\frac{\eta {}^{\prime }-\xi }{\lambda }\right),\;\left(\eta {}^{\prime }>\xi \right)$$

(3) Bounded System *S_B_*:(3)$$\eta =\gamma +\delta \;\log\left(\frac{\eta {}^{\prime }-\xi }{\xi +\lambda -\eta {}^{\prime }}\right),\;\left(\eta {}^{\prime }>\xi \right)$$
where *γ*, *δ*, *ξ* and *λ* are the four conversion coefficients calculated from a given skewness and kurtosis, *η* is a random sequence of the standard normal distribution, and *η*′ is a non-Gaussian sequence with a given skewness and kurtosis.

The steps to simulate a non-Gaussian rough surface are as follows:

(1) Generate a discrete Gaussian random sequence *η* (*m*, *n*) from a random number generator for input;

(2) The autocorrelation function for a simulated surface is:(4)$$R\left(x,y\right)=\sigma^{2}\exp\left\{-2.3{\left[{\left(\frac{x}{\beta_{x}}\right)}^{2}+{\left(\frac{y}{\beta_{y}}\right)}^{2}\right]}^{\frac{1}{2}}\right\}$$
where *β_x_* and *β_y_* are the autocorrelation lengths of the surface profile in the *x* and *y* directions, respectively;

(3) The Fourier transform of the autocorrelation function determines the probability density function of the output sequence:

where *u* = 0, 1, 2, …, *m* − 1 and *v* = 0, 1, 2, …, *n* − 1;

(4) The probability density function of the input sequence is a constant which is set to *S_η_*(*u*, *v*), then the transfer function of the system is
(5)$$H\left(u,v\right)={\left[S_{z}\left(u,v\right)/S_{\eta }\left(u,v\right)\right]}^{\frac{1}{2}}$$

(5) Perform the Fourier inverse transform on the transfer function to find the filter function:(6)$$h\left(x,y\right)=\frac{1}{mn}\sum_{x=0}^{m-1}\sum_{y=0}^{n-1}H\left(u,v\right)\exp\left(\frac{j2\pi ux}{m}\right)\exp\left(\frac{j2\pi uy}{n}\right)$$

(6) Convert the Gaussian distribution sequence into a non-Gaussian sequence of positive skewness and kurtosis by using the Johnson conversion system;

(7) Filter the non-Gaussian sequence with the filter function *h(x, y)* to obtain a random output sequence *z(m, n)* with a specific distribution;
(7)$$z\left(m,n\right)=\sum_{x=0}^{m-1}\sum_{y=0}^{n-1}h\left(x,y\right)\eta \left(m-x,n-y\right)$$

A non-Gaussian rough surface was generated, as shown in Figure 1, wherein the given skewness was 0.2, the kurtosis was 3, and the autocorrelation lengths in the *x* and *y* directions were both 1.

### 2.2. Construction of Texture Model

The surface roughness of the microgrooves is affected by both the feed rate and the spacing of adjacent grooves at the same time [22,23]. Figure 2 shows the digital process of microgroove by the ball-end milling cutter with two straight lutes. The coordinate system was established with the point O as the coordinate origin, and the following formula can obtain the point *S* (*x*, *y*, *z*) of the surface of the microgroove:(8)$$S=\left\{\begin{array}{c} x\\ y\\ z \end{array}\right\}=\left\{\begin{array}{c} nP+R\cos\beta \cos\left(\alpha +\omega t\right)\\ v_{f}t+R\cos\beta \sin\left(\alpha +\omega t\right)\\ R\sin\beta \end{array}\right\}$$
where *n* is the number of pick-paths, *P* is the chosen path or considered path, *R* is the radius of the ball end milling cutter, ω is the spindle speed, *t* is the processing time, *vf* is the feed rate, *α* is the angle between a line and the *x*-axis, *α* is the angle between the *x*-axis and the perpendicular from the considered point to axis Oz, while *β* is the angle between the locus vector S and the plane xOy.

This paper established a periodic regular texture model for the machining parameters of the micromilling process, as shown in Figure 3, where the spindle speed was 10,000 r/min, the feed rate was 200 μm/min, and the spacing of adjacent grooves was 100 μm.

### 2.3. Synthesis of Microgroove Surface

Referring to the actual surface micromorphology of micromilling in Section 4.2, the scale factor of the simulated non-Gaussian rough surface was 0.3. Then, the simulated proportional non-Gaussian rough surface was added to the texture model. The synthesized microgroove surface *Z_H_* is shown in Figure 4. The calculation formula is
(9)$$Z_{H}=\lambda \backslash cdotZ_{C}+Z_{W}$$
where *λ* is the scale factor, *Z_C_* is the non-Gaussian rough surface, and *Z_W_* is the texture model.

## 3. Determination of BEMD Reference

In surface micromorphology, the wave distance of roughness is the shortest, which is a high-frequency signal, and the surface waviness and surface shape error have relatively long wave distances, which are low-frequency signals. In this paper, we arranged the height information of the micromilling surface in a matrix and converted it into a two-dimensional image signal. Using the BEMD algorithm, the microgroove surface can be adaptively decomposed; then, the high-frequency signal and the low-frequency signal were separated to extract the evaluation data of its three-dimensional surface roughness [24,25].

### 3.1. Theory of BEMD

The conditions for signal decomposition are that there is at least one maximum point and minimum point in the signal, or the whole signal has no extreme point but is derived by a first or several orders operation to obtain a maximum point and a minimum point.

The specific process of BEMD to decompose signal is as follows:

(1) The two-dimensional digital sequence and the residual component of the signal are *g*(*x*, *y*) and *r_i_*(*x*, *y*) (*i* = 0, 1, 2, …), respectively, where *r*_0_(*x*, *y*) = *g*(*x*, *y*);

(2) Find all the maximum and minimum points of *r_i_* (*x*, *y*) for interpolation, fit the upper envelope surface *U_i_*(*x, y*) and lower envelope surface *D_i_*(*x, y*), and take the mean value:(10)$$E_{i}\left(x,y\right)=\frac{U_{i}\left(x,y\right)+D_{i}\left(x,y\right)}{2}$$

(3) Judge whether *r_i_*_+1_(*x*, *y*) = *r_i_*(*x*, *y*) − *E_i_*(*x*, *y*) satisfies the termination condition of BIMF; if it is satisfied, BIMF*_i_*(*x*, *y*) = *r_i_*_+1_(*x*, *y*), if not, then let k = k + 1, and go to step (2);

(4) Repeat steps (2) and (3) until all BIMFs and the residual are obtained.

The discriminant function of termination for screening is
(11)$$SD=\sum \frac{{\left[r_{i}\left(x,y\right)-r_{i+1}\left(x,y\right)\right]}^{2}}{r_{i}{\left(x,y\right)}^{2}}$$

The value of SD is generally between 0.2 and 0.3, and this paper has selected 0.2.

### 3.2. Inhibition of Boundary Effect of BEMD

In the actual process of decomposition, the marginal data of the image are not necessarily the maximum value or the minimum value. During the envelope fitting, the marginal data are interpolated continuously, which causes the divergence of the marginal data and the continuous diffusion into the data, resulting in the boundary effect of BEMD. We selected a simple and effective method called mirror extension [26]. Theoretically, the boundary effect cannot be eliminated entirely but can only minimize the influence of the boundary effect. Figure 5 compares the raw data and the two-dimensional cross section of a BIMF in the x and y directions after processing by the mirror method. The results show that the boundary effect is significantly suppressed by the image method.

### 3.3. Establishment of the Reference Plane

The surface morphology of micromilling consists of shape error, surface waviness, and roughness [27], where the roughness is a high-frequency signal. Let the three-dimensional surface profile be *z*(*x*, *y*), the roughness signal be *r*(*x*, *y*), and the low-frequency reference signal be *w*(*x*, *y*), where *w*(*x*, *y*) includes shape error and surface waviness. The mathematical model for the separation of surface roughness is:(12)$$z\left(x,y\right)=r\left(x,y\right)+w\left(x,y\right)$$

The BEMD algorithm was used to decompose the simulated microgroove surface after mirror extension, and the fast Fourier transform was performed on the decomposed BIMFs and the residual to obtain the time domain and frequency domain diagrams, as shown in Figure 6 and Figure 7, respectively. Figure 8 shows the energy weight and correlation of each BIMF and the residual with the original image. The energy calculation is shown in Formula (14), which is the square mode of the energy spectral density after the Fourier transform of the signal. Let the two images be *A*(*x*, *y*) and *B*(*x*, *y*), respectively, then the calculation method of the correlation coefficient is as shown in Formula (15).
(13)$$F\left(x,y\right)={\left|\sum_{x=0}^{m-1}\sum_{y=0}^{n-1}z\left(x,y\right)e^{-j2\pi \left(\frac{ux}{m}+\frac{vy}{n}\right)}\right|}^{2}$$
where *u* = 0, 1, 2, …, *m* − 1 and *v* = 0, 1, 2, …, *n* − 1.
(14)$$R\left(x,y\right)=\frac{\sum_{x=0}^{m-1}\sum_{y=0}^{n-1}A\left(x,y\right)B\left(x,y\right)}{{\left[\sum_{x=0}^{m-1}\sum_{y=0}^{n-1}A\left(x,y\right)\right]}^{\frac{1}{2}}{\left[\sum_{x=0}^{m-1}\sum_{y=0}^{n-1}B\left(x,y\right)\right]}^{\frac{1}{2}}}$$

Micromilling surface topography information is a signal with a high signal-to-noise ratio. In Figure 6, from the perspective of the time domain, the surfaces of BIMF1 and BIMF2 have visible micromorphology of a rough surface. In Figure 7, from the aspect of the frequency domain, the roughness information is the high-frequency signal. In Figure 8, from an energy point of view, in general, the energy of the noise decreases as the number of decomposition layers increases until the first minimum point occurs, and then the energy of microgroove signals dominates. At the same time, the similarity between the high-frequency noise and the original signal is low, while the similarity between the low-frequency signal and the original image is high. Therefore, the number of decomposition layers at which the energy curve and the cross-correlation coefficient curve begin to increase for the first time is taken as the critical point of signal-to-noise mode. Furthermore, the energy of the BIMF2 and the correlation coefficient between the BIMF2 and the original image become the first point from descent to ascent at the same time. Thus, BIMF1 and BIMF2 were taken as surface roughness for reconstruction, and the remaining components were the reference for evaluation. The reconstructed surface is shown in Figure 9.

### 3.4. Errors Analyses of the Reference Plane

Because of the influence of modal aliasing, the reconstructed rough surface still retains the signal of the reference plane. The texture of the microgroove can be seen in Figure 9a, and the mean squared error (MSE) of the reference plane is 1.03%. Among many filters, Butterworth low-pass filter has the advantages of balanced characteristics in three aspects: linear phase, attenuation slope, and loading characteristics. The rough surface, therefore, was filtered by a Butterworth third-order low-pass filter and added to the reference plane to reconstruct a new reference plane [28], and the amplitude squared response of the filter is shown in Formula (16). The new rough surface and the reference plane are shown in Figure 10. The microgroove texture of the rough surface is substantially invisible, while the reference plane is smoother than the reference plane without filtering, and the MSE has reduced to 0.51%. Table 1 shows the MSE of the reference plane of the simulated non-Gaussian rough surface under different influencing parameters, where bx and by are the autocorrelation length in the x and y directions, respectively. Table 2 shows the MSE of the reference plane under different micromilling parameters. When the influence parameters and processing parameters change, the fixed parameter values were selected from the following values: the autocorrelation lengths in both x and y directions were both 1, kurtosis was 3, skewness was 0.2, spindle speed was 10,000 r/min, the feed rate was 200 μm/min, and spacing of adjacent grooves was 100 μm. Errors of the measured surfaces are below 1%, indicating that the method can effectively extract the reference plane of the three-dimensional surface roughness of the microgroove surface. Figure 11 is a flow chart for extracting the reference plane.
(15)$${\left|H\left(j\omega \right)\right|}^{2}=\frac{1}{1+{\left(\omega /\omega_{c}\right)}^{2n}}$$
where *ω* is the frequency of the input signal, *ω_c_* is the cutoff frequency, and *n* is the order of the filter.

## 4. Experimental Analyses

### 4.1. Experimental Equipment and Method

In this paper, the experiment was conducted using a vertical machining center (Hass). A double-edged ball-end milling cutter with a diameter of 1 mm was selected, and the processing material was aluminum alloy (7075). Processing parameters are shown in Table 3. The machined surface was measured using a laser confocal microscope (LMS700). Actual surfaces are shown in Figure 12.

### 4.2. Analyses of Experimental Results

In the traditional milling process, the surface roughness along the center line of the surface is usually used as an evaluation index for evaluating the roughness of the milling surface. The conventional milling surface roughness value is approximately proportional to the square of the single tooth feed. This standard applies when the tool is in good condition and the surface forming path is clear. However, micromilling is different from conventional milling. Because of the large reduction in the size of the tool and workpiece, the size effect is generated. That is, as the thickness of the cutting decreases, the unit cutting force required to form the chip increases nonlinearly, and the smaller the thickness, the more obvious the unit cutting force increases [28,29,30,31,32]. Because of the plastic deformation of the material after processing, microburrs are generated when the cutting edge leaves the surface to be machined, which is a significant factor affecting micromilling. At the same time, during the micromilling process, the minimum effective cutting thickness which can stabilize the cutting is called the minimum cutting thickness, and the minimum cutting thickness is also the main factor affecting the quality of the machined surface. When the cutting thickness is less than the minimum cutting thickness, a plowing effect is generated, which in turn leads to an increase in the cutting force and the surface roughness of the machined surface. Therefore, the tool path of the machined surface is variable and no longer clear. At this time, if the surface roughness along the center line is not used to describe the surface quality accurately, the surface roughness of the entire surface is used to evaluate the milled surface.

The parameters of three-dimensional surface roughness for evaluation spatially reflect the overall surface characteristics, which overcome the locality of two-dimensional evaluation. Excessive characteristic parameters, however, are likely to cause an “explosion of parameters”, and it is meaningless, so several representative parameters, according to the structural characteristics of the microgroove surface, ISO-25178 standard, are selected to characterize the microgroove surface [33,34,35], as shown in Table 4, where *Sa*, *Sq*, *Ssk*, *Sku*, and *Sz* are amplitude parameters; *Sdq* and *Sdr* are hybrid parameters. We used LMS700 to derive three-dimensional data of the actual microgroove surfaces and calculated the three-dimensional surface roughness on the basis of the reference plane extracted by BEMD. Figure 13 shows the two-dimensional surface roughness, and Figure 14 shows the three-dimensional surface roughness. Since the numerical range differs significantly, it was divided into two graphs for observation.

By comparing Figure 13 and Figure 14, the corresponding order of roughness parameters is: *Ra* is equivalent with *Sa*, *Rku* equivalent with *Sku*, *Rq* equivalent with *Sq*, *Rsk* equivalent with *Ssk*, *Rz* equivalent with *Sz*, it can be seen that the variation trends of each two-dimensional surface roughness parameter and its corresponding three-dimensional surface roughness parameter are inconsistent. The calculation of two-dimensional surface roughness is performed only for the roughness of an individual line on the surface and cannot accurately judge the difference between different surfaces, while for three-dimensional surface roughness, it takes into account the whole surface. As a result, two-dimensional surface roughness parameters cannot accurately describe the characteristics of the surface. We analyzed the variation trend of three-dimensional surface roughness.

The spindle speed shown on surface 2 decreases compared with that on surface 1. Unlike conventional milling, the blunt radius remains unchanged, and the cutting thickness of the material to be removed is much smaller in micromilling operations. The ratio of the blunt radius of the cutting edge of the tool to the cutting thickness is significantly reduced, resulting in relatively high cutting force and vibration. The increase in the spindle speed will exacerbate the chatter, leading to poorer surface quality. When the spindle rotates at high speed, the radial runout of the spindle is amplified because of the increase in the centrifugal force, exerting an effect on the bending deformation of the tool. The bending deformation of the tool decreases as the spindle speed increases.

Meanwhile, feed per tooth changes with an increase or decrease in spindle speed, which affects the residual height of the surface. The spindle speed and surface roughness, therefore, are not linear relationships. Results of experiments show that the values of *Sa* and *Sq* of surface 2 increased compared with those of surface 1, wherein *Sa* is the extension of the parameter of two-dimensional surface roughness in three dimensions and reflects the deviation degree of the surface profile from reference together with *Sq*. The skewness *Ssk* measures the symmetry of the surface deviation concerning the reference plane, and *Sdq* represents the root mean square value of the slope of all points on the measured surface. It is observed that the feed per tooth of surface 2 increases, and the residual height of the groove bottom increases compared with that of surface 1; therefore, the absolute value of skewness is more significant than that of surface 1. Moreover, the surface area of the groove bottom becomes larger, but the density of the burr witnesses a slight increase, so there is no noticeable change in the value of *Sdq* compared with surface 1. *Sz* is an amplitude parameter and is used to indicate the limit of the distance between peak and valley heights in the evaluation area. *Sdr* is the area ratio of surface contact, and *Sku* reflects the height distribution of the surface. Decreasing spindle speed causes the burr height on the top of the groove to decrease, resulting in a decrease in a flutter. Considering that the overall change of the burr height on the surface is relatively slow, *Sz*, *Sdr*, and *Sku* represent a decrease accordingly.

In contrast to surface 1, the feed rate of surface 3 was set at a higher value, thus, leading to an increase in feed per tooth. Consequently, the cutting force is increased, tool deformation becomes worse, and the residual height of the surface becomes increased, resulting in an increase in the deviation degree of the surface profile and larger values of *Sa*, *Sq*, and *Sdr* compared with those of surface 1. The small cut thickness in the process of micromilling causes the extrusion and friction between the tool and chip. As the feed per tooth increases, the burr size decreases; consequently, *Sz* and *Sku* both decrease because of the lowered kurtosis. The feed per tooth of surface 3 relative to surface 1 is larger than that of surface 2 relative to surface 1, and the tool marks on the bottom surface of the groove are sparse so that the height and density of the bottom burrs of the groove are larger than that of the surface 2, and *Sdq* is larger than that of the surface 2. However, the absolute value of *Ssk* is between those of surface 1 and surface 2 because of the larger increase in *Sq*.

The spacing of grooves of surface 4 is increased compared with surface 1, i.e., the overlap rate of the tool mark decreases. The cutting thickness of the micromilling and the groove width increase. Although the surface area of the microgroove is increased, the area of the bottom of the groove is smaller than the whole surface. The burr height on the top of the groove caused by the extrusion of the tool is reduced, the overall height distribution of the burr is relatively uniform, and the degree of deviation of the surface contour is not apparent, so *Sa* increases and Sq does not change significantly, while *Sz* decreases drastically. In addition, the separated rough surface is less affected by the groove texture, and the absolute value of *Ssk* decreases. An increase in the surface area of the microgroove leads to an increase in the density of the burr so that the values of *Sdq* and *Sdr* are significantly increased. *Sku* increases because the burr becomes sharper.

The five amplitude parameters selected in this paper can analyze the surface height distribution from different angles, including the difference in residual altitude, the deviation of the surface profile from the reference plane, and the variation of the residual height. The two-hybrid parameters reflect the surface characteristics from both the height and the spacing behaviors. Seven parameters, therefore, can more fully characterize the three-dimensional surface roughness of the curved surfaces of the microgrooves so that the four surfaces can be easily distinguished by quantification.

## 5. Conclusions

In this paper, micromilling microgroove models were established, and the reference surfaces of the three-dimensional surface roughness were extracted to establish an evaluation system of three-dimensional surface roughness. The main conclusions are as follows:

1. The microgroove surfaces under different processing parameters can be proportionally synthesized by texture models and non-Gaussian random rough surfaces, which are simulated by the fast Fourier transform and the Johnson transformation system;

2. The boundary effect of the BEMD can be suppressed by mirror extension;

3. According to the time domain, frequency domain, energy, and cross-correlation coefficient of each BIMF, the surface roughness of the microgroove can be separated, and the reference plane of the three-dimensional surface roughness can be effectively extracted;

4. Through error analyses, it is proved that BEMD can effectively extract the reference plane of the microgroove so as to analyze the surface roughness of the microgroove;

5. Using the BEMD algorithm to characterize the curved surface topography of the micromilling microgroove under different processing parameters, it is verified that the three-dimensional roughness evaluation system composed of seven parameters selected from ISO-25178 standard can characterize different microgroove surfaces.

## Figures and Tables

**Figure 1 micromachines-13-02011-f001:**
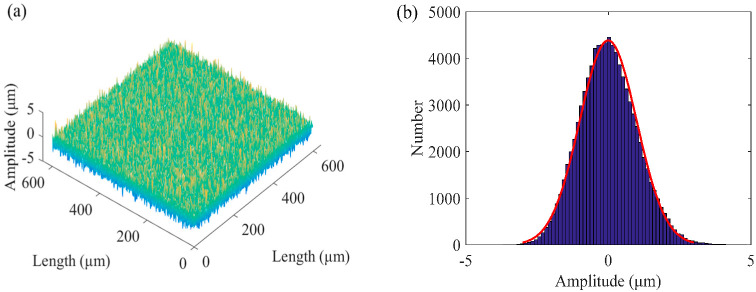
A simulated non-Gaussian rough surface: (**a**) non-Gaussian rough surface; (**b**) histogram of the non-Gaussian rough surface.

**Figure 2 micromachines-13-02011-f002:**
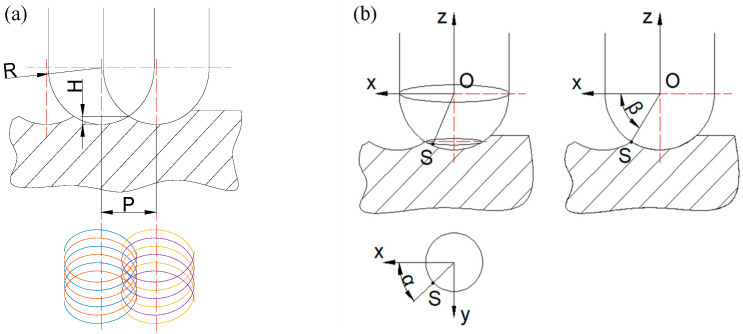
The model of the resulting microgroove: (**a**) the locus of a ball-end milling cutter with two straight flutes. (**b**) coordinate of the locus of the ball-end milling.

**Figure 3 micromachines-13-02011-f003:**
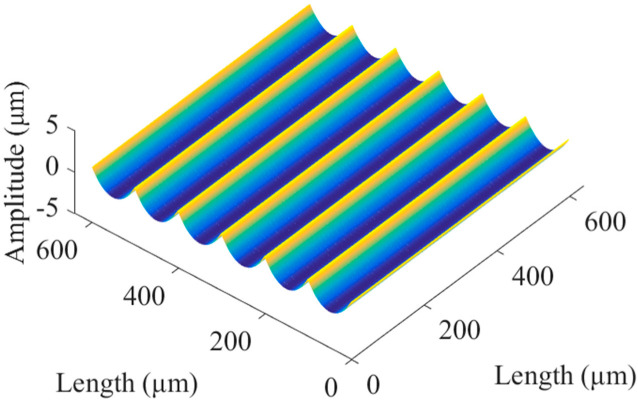
Texture model.

**Figure 4 micromachines-13-02011-f004:**
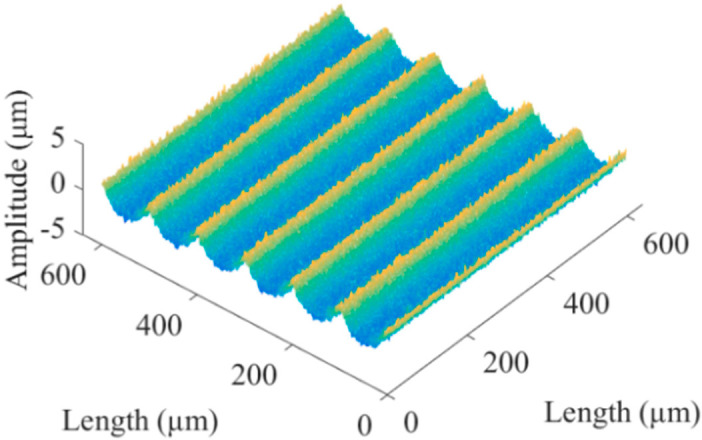
Simulated microgroove surface.

**Figure 5 micromachines-13-02011-f005:**
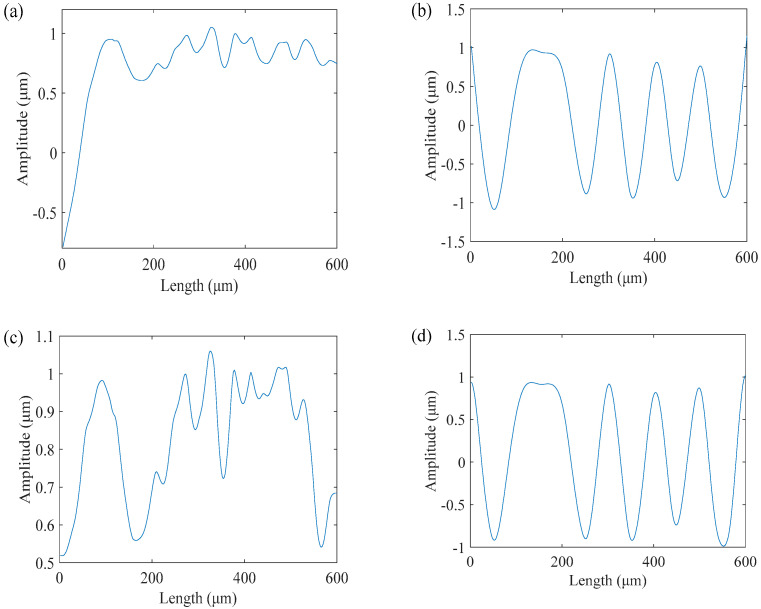
Comparison of a BIMF before and after processing: (**a**) 2D cross section of a BIMF of raw data in x direction; (**b**) 2D cross section of a BIMF of raw data in y direction; (**c**) 2D cross section of a BIMF in x direction after processing; (**d**) 2D cross section of a BIMF in y direction after processing.

**Figure 6 micromachines-13-02011-f006:**
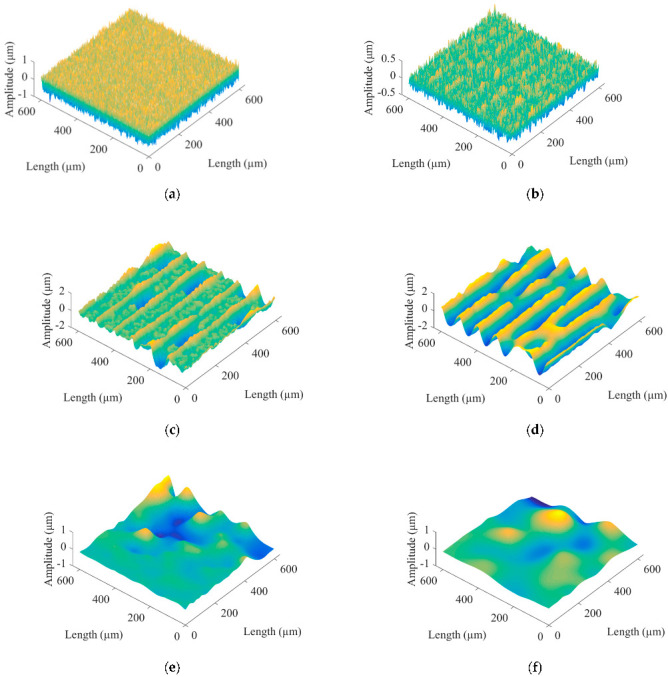
Time domain diagrams of BIMFs and residual. (**a**) BIMF1. (**b**) BIMF2. (**c**) BIMF3. (**d**) BIMF4. (**e**) BIMF5. (**f**) BIMF6. (**g**) BIMF7. (**h**) residual.

**Figure 7 micromachines-13-02011-f007:**
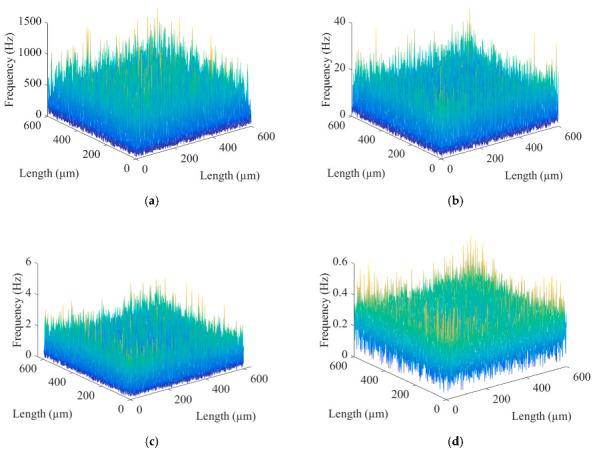
Frequency domain diagrams of BIMFs and the residual: (**a**) BIMF1; (**b**) BIMF2; (**c**) BIMF3; (**d**) BIMF4; (**e**) BIMF5; (**f**) BIMF6; (**g**) BIMF7; (**h**) residual.

**Figure 8 micromachines-13-02011-f008:**
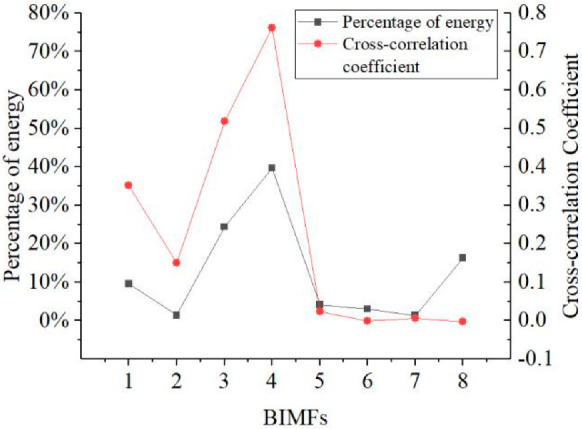
Energy weight and correlation of each BIMF and residual with the original image.

**Figure 9 micromachines-13-02011-f009:**
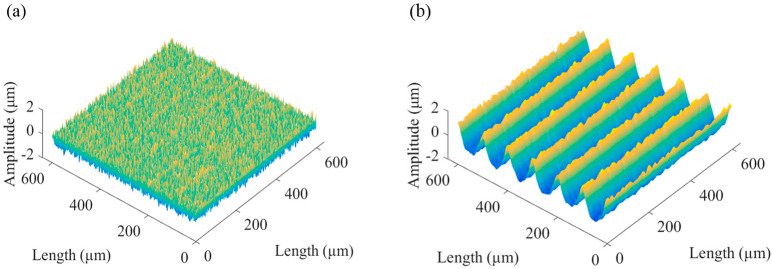
Reconstructed surfaces: (**a**) reconstructed rough surface; (**b**) reference plane.

**Figure 10 micromachines-13-02011-f010:**
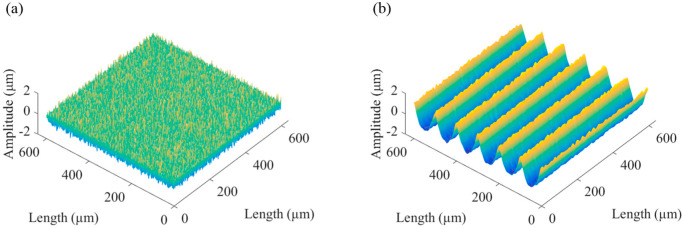
Rough surface and reference plane reconstructed after filtering: (**a**) reference plane; (**b**) reconstructed rough surface.

**Figure 11 micromachines-13-02011-f011:**
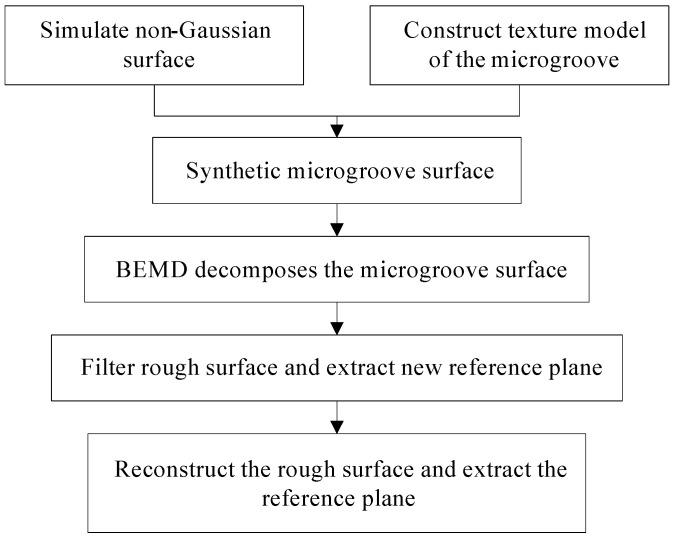
The extraction process of the reference plane.

**Figure 12 micromachines-13-02011-f012:**
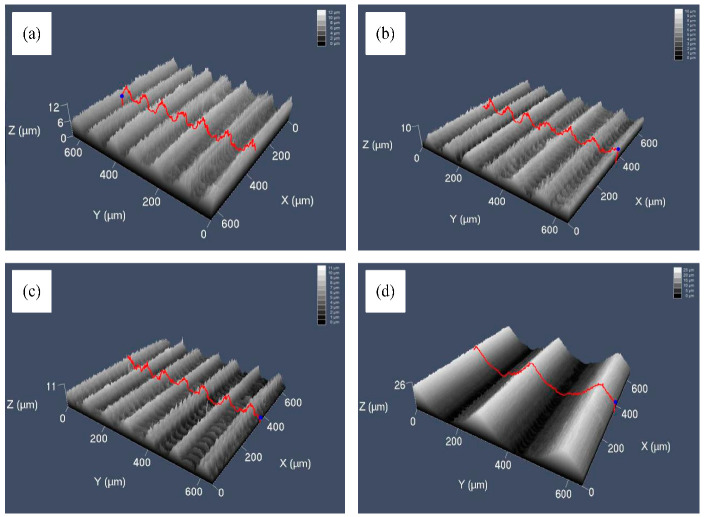
Actual microgroove surfaces: (**a**) surface 1; (**b**) surface 2; (**c**) surface 3; (**d**) surface 4.

**Figure 13 micromachines-13-02011-f013:**
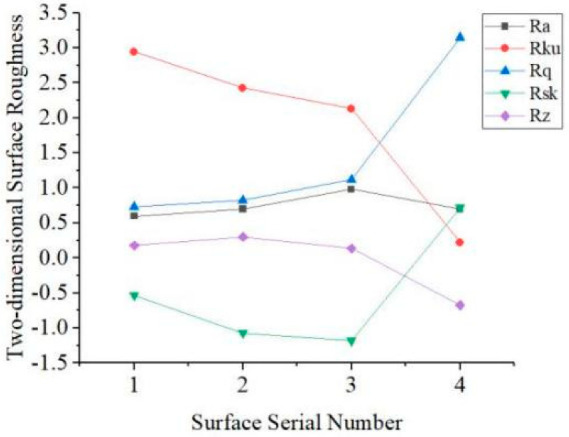
Two-dimensional surface roughness of microgroove surfaces.

**Figure 14 micromachines-13-02011-f014:**
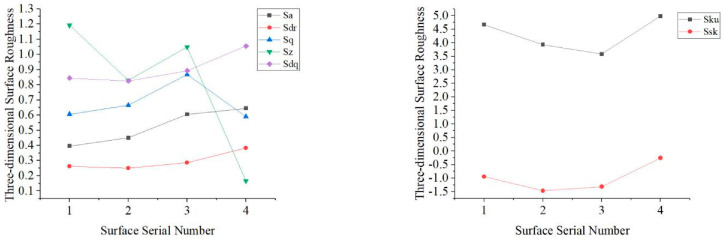
Three-dimensional surface roughness of microgroove surfaces.

**Table 1 micromachines-13-02011-t001:** MSE of reference plane at different rough surfaces.

Level	Influence Factors							
	Isotropy	MSE	Anisotropy	MSE	Kurtosis	MSE	Skewness	MSE
1	*bx* = 0.1 *by* = 0.1	0.44%	*bx* = 1 *by* = 0.1	0.46%	2.5	0.48%	−0.2	0.50%
2	*bx* = 0.5 *by* = 0.5	0.45%	*bx* = 1 *by* = 0.5	0.47%	3	0.51%	0.2	0.51%
3	*bx* = 1 *by* = 1	0.51%	*bx* = 0.5 *by* = 0.1	0.44%	3.5	0.52%	0.4	0.49%

**Table 2 micromachines-13-02011-t002:** MSE of the reference plane under different micromilling parameters.

Level	Processing Parameters					
	Spindle Speed (r/min)	MSE	Feed Rate (μm/min)	MSE	Spacing of Grooves (μm)	MSE
1	8000	0.52%	100	0.50%	100	0.51%
2	9000	0.50%	200	0.51%	200	0.70%
3	10,000	0.51%	300	0.52%	300	0.92%

**Table 3 micromachines-13-02011-t003:** Machining parameters of microgrooves.

Number	Processing Parameters		
	Spindle Speed (r/min)	Feed Rate (μm/min)	Spacing of Grooves (μm)
1	10,000	200	100
2	7000	200	100
3	10,000	300	100
4	10,000	200	300

**Table 4 micromachines-13-02011-t004:** Three-dimensional surface roughness parameters.

Parameters	The Formula for Calculating Parameters
Arithmetic mean deviation	$S_{a}=\frac{1}{A}\iint_{A}\left|z\left(x,y\right)\right|dxdy$
Root mean square deviation	$S_{q}=\sqrt{\frac{1}{A}\iint_{A}z^{2}\left(x,y\right)dxdy}$
Skewness	$S_{sk}=\frac{1}{S_{q}{}^{3}}\left[\frac{1}{A}\iint_{A}z^{3}\left(x,y\right)dxdy\right]$
Kurtosis	$S_{ku}=\frac{1}{S_{q}{}^{4}}\left[\frac{1}{A}\iint_{A}z^{4}\left(x,y\right)dxdy\right]$
Averaged peak to valley	$S_{z}=\frac{1}{25}\left[\overset{5\sum \sum_{i=1}^{5\sum }z_{i}{\left(x,y\right)}_{min}}{\underset{i=1}{\sum }}z_{i}{\left(x,y\right)}_{max}\right]$
Developed surface area ratio	$S_{dq}=\sqrt{\frac{1}{A}\iint_{A}\left[{\left(\frac{\partial z\left(x,y\right)}{\partial x}\right)}^{2}+{\left(\frac{\partial z\left(x,y\right)}{\partial y}\right)}^{2}\right]dxdy}$
Area ratio of surface contact	$S_{dr}=\frac{1}{A}\left[\iint_{A}\left(\sqrt{\left[1+{\left(\frac{\partial z\left(x,y\right)}{\partial x}\right)}^{2}+{\left(\frac{\partial z\left(x,y\right)}{\partial y}\right)}^{2}\right]}-1\right)dxdy\right]$

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
