# Peer review of "Evaluation of Three-Dimensional Surface Roughness in Microgroove Based on Bidimensional Empirical Mode Decomposition"

_micromachines, 2022, doi:10.3390/mi13112011_

Round 1

Reviewer 1 Report

(1) Increase the number of the reference papers including (primarily) from MDPI journals.

(2) The authors must improve the Abstract.The description is too long and can be refined.

(3) The author must give a detailed description of the seven milling parameters(Sa, Sq, Ssk, Sku,Sz,Sdq,Sdr),and explain why the seven parameters are selected to evaluate the three-dimensional surface roughness.

Author Response

Dear Editor,

Thanks very much for the reviewer’s constructive comments. It helped us to improve the manuscript significantly. Each comment of the reviewer is followed by our reply; corresponding changes to the manuscript are highlighted in red.

With best regards,

The authors

----------- Comments by Reviewer #1-----------

Comment 1

Increase the number of the reference papers including (primarily) from MDPI journals.

Reply: Thanks for the reviewer’s suggestion, we have added three references from MDPI journals.

Comment 2

The authors must improve the Abstract. The description is too long and can be refined.

Reply: Thanks for the reviewer’s suggestion. The Abstract have already been refined in the revised file. They are highlighted in red.

Comment 3

The author must give a detailed description of the seven milling parameters(Sa, Sq, Ssk, Sku,Sz,Sdq,Sdr),and explain why the seven parameters are selected to evaluate the three-dimensional surface roughness.

Reply:

Sa is the arithmetic mean height of the surface, which belongs to the amplitude parameter. Represents the arithmetic mean of the absolute value of the surface height.

Sq is the surface root mean square height, which belongs to the amplitude parameter. The measured value is the root mean square of the surface height distribution value.

Ssk is the surface deviation, which belongs to the amplitude parameter. It describes the measurement of the symmetry of the surface height distribution relative to the datum plane. The deflection of the Gaussian rough surface is 0. If Ssk>0, it means that the peak value of the surface is more than the valley value. On the contrary, if Ssk<0, it means that the valley value of the surface is more than the peak value.

Sku is the surface kurtosis, which belongs to the amplitude parameter. The kurtosis of Gaussian rough surface is 3. When Sku>3, the peak and valley of the surface are more than those of Gaussian surface; When Sku<3, the peaks and valleys of the surface are less than those of Gaussian surface.

Sdq The slope of root mean square of the surface is a mixed parameter, which refers to the root mean square of the surface gradient in the assessment area.

Sdr Is the surface expansion area ratio, which belongs to a mixed parameter and represents the ratio of the interface area in the assessment area to the increment of the assessment surface area.All parameters are described in Line377-line 439.

The reason why the seven parameters are selected to evaluate the three-dimensional surface roughness is as follows:

The draft roughness standard ISO/TC213N756 divides the three-dimensional roughness parameters into five groups: height parameters, volume parameters, functional parameters, mixed parameters and spatial parameters, totaling 23 parameters.However, if all the 23 parameters are used to represent the roughness of the three-dimensional plane, this will produce parameter explosion effect and the calculation will be complicated. Therefore, we have done a lot of experimental work in the early stage, and selected 7 representative parameters to roughness of the three-dimensional plane.

Reviewer 2 Report

Congratulations for this, mathematically well supported and valorous conclusions containing article.

See the attachment below containig my suggestions and comments.
